# Silicon Steel Strip Profile Control Technology for Six-High Cold Rolling Mill with Small Work Roll Radius

**Hainan He, Jian Shao \*, Xiaochen Wang, Quan Yang and Xiawei Feng**

National Engineering Research Center for Advanced Rolling Technology, University of Science and Technology Beijing, Beijing100083, China; csuhhn@163.com (H.H.); xcwangustb@163.com (X.W.); yangquan@nercar.ustb.edu.cn (Q.Y.); fxw@tc124.com (X.F.)

\* Correspondence: jianshao@ustb.edu.cn; Tel.: +86-10-62336320

**Abstract:** Due to the requirement of magnetic properties of silicon steel sheets, producing high-precision size strips is the main aim of the cold rolling industry. The tapered work roll shifting technique of the six-high cold rolling mill is effective in reducing the difference in transverse thickness of the strip edge, but the effective area is limited, especially for a high crown strip after the hot rolling process. The six-high mill with a small work roll size can produce a strip with higher strength and lower thickness under a smaller rolling load. At the same time, the profile of the strip can be substantially improved. By advancing a well-established analytical method, a series of simulation analyses are conducted to reveal the effectiveness of a small work roll radius for the strip profile in the six-high cold rolling process. Through the analysis of flattening deformation and deflection deformation on the load, the change rule of the strip profile produced by the work roll with a small roll diameter can be obtained. Combined with theoretical analysis and industrial experiments, it can be found that the improvement effect of the small work roll radius on the profile of the silicon strip is as significant.

**Keywords:** small work roll radius; cold rolling; silicon steel strip profile; roll deformation model

## 1. Introduction

As a functional material commonly found in magnetic cores, producing a flat thin silicon steel strip is imperative to ensure its functionality when being stacked into cores. The cold rolling process has a significant influence on the grain size and texture of non-oriented silicon steel, and affects the iron loss and magnetic properties of a silicon steel sheet. In addition, the thickness consistency of a cold-rolled strip also affects the magnetic properties of a silicon steel sheet. As the silicon steel sheet is used in lamination, Xia et al. [1] found that a 1% decrease in the lamination factor leads to a 2% increase in iron loss and a 1% decrease in induction. Increasing the lamination factor requires a silicon steel strip with a good profile, i.e., a slight transverse thickness difference along the strip width. In the rolling process of a silicon steel strip, it is extremely important to improve the transverse thickness accuracy of the strip while maintaining the thickness accuracy along the rolling direction. Edge drop control technology of a hot- and cold-rolled strip is an important means to improve the transverse thickness accuracy of the strip.

Many researchers have focused on improving the non-silicon strip profile during hot rolling in recent years. Yao et al. [2] developed a specified tapered work roll shifting strategy in four-high hot rolling for silicon strips with the same width. Ma et al. [3] developed a large concave roll for a four-high hot rolling process so that the wedge of the strip can be reduced. However, the hot rolling thin strip is still a complicated production system by Zhao et al. [4], and there is room to improve to directly produce 0.3–0.5 mm thick thin silicon strips.

Cold rolling remains indispensable to reducing the hot-rolled 2.2 mm strip into a product with superior surface and profile quality. However, because silicon steel becomes harder under low temperature, an excessively larger pressure is needed to plastically deform the thin silicon strip. Given the fact that the roller has to be wider than the strip width and is made of steel, the difference of work roll elastic deformation is larger across the strip width especially at the strip edge drop area. A larger work roll flattening difference leads to a larger strip thickness difference and larger trimming width, and a consequently lower production competitiveness in magnetic and electrical devices. Wang et al. [5] mentioned that grinding the work roll into a tapered contour is the most effective way to compensate for the strip thickness difference.

Numerous tapered work roll contours are designed according to the control of the strip profile. Ma et al. [6] developed a symmetrical tapered work roll contour with unknown tapered height, and claimed that the controlling width was 100 mm. It was designed for a six-high reversal mill with a work roll diameter of 240 mm. Cao et al. [7] proposed a single tapered 0.8 mm × 120 mm (height × length) contour for a four-high tandem rolling mill, in which the work roll diameter was 600 mm. Sun et al. [8] developed a work roll contour for the first stand of a six-high tandem rolling process. Liu et al. [9] developed a 0.5 mm × 155 mm (height × length) contour for the fifth stand of a six-high tandem rolling process. Zhang et al. [10] proposed a 0.9 mm × 155 mm contour for the first stand of a 1500 mm six-high tandem rolling line. Zhang et al. [11] proposed a 0.98 mm × 205 mm single tapered contour for the first and second stands of a six-high tandem rolling line with a 425 mm work roll diameter. All the above-mentioned contours with noteworthy differences were implemented and reported effective in improving the strip profile of a silicon steel strip. The question that needs to be answered is, for the same silicon steel strip, why does the work roll contour differ so significantly?

Most recently, the authors enjoyed the privilege of partnering with a silicon steel strip manufacturer with the purpose of improving a non-oriented silicon steel (Si < 3 wt.%) strip profile. The producer owns a pallet consisting of a six-high reversal mill shown in Figure 1 with a larger work roll diameter. The six-high mill is the most commonly used equipment for cold rolling of silicon steel. The cause of the strip edge-drop is the flattening deformation of the roll. Many of the above tapered work roll contours are used to improve the edge profile of the strip, but the effect is limited. In cold rolling mill equipment, a high-strength steel strip can be produced using mills equipped with more rolls (such as 12 high and 20 high mills) with the smaller work roll diameter, which can not only reduce the rolling load but also increase the stability of the roll system to ensure the shape quality of the strip profile. Based on the existing rolls system structure of the rolling mill (six-high reversal mill), the possibility of reducing the work roll diameter was explored to improve the equipment's ability to produce a non-oriented silicon steel strip with a higher silicon content and better thickness accuracy, and, at the same time, the equipment transformation cost was relatively low. In addition, the ability of the tapered work roll to control the strip edge shape under different work roll diameters also needs to be analyzed and studied. It has become the most effective tool for producing a high-grade silicon steel strip compared to other rolling mill configurations.

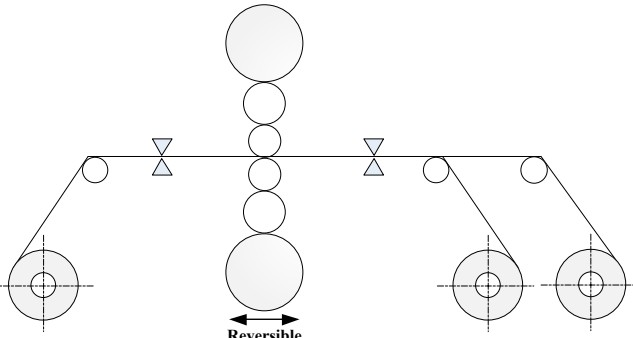

**Figure 1.** Rolling diagram of six-high reversing cold mill.

The rolling process of a strip in a rolling mill is a complicated deformation process. Analysis methods and models with low computational cost and high enough accuracy need to be taken into account when analyzing the rolling mill-strip deformation behavior. The model separates the roll stacks and strip steel and applies appropriate conditions for analysis, and the results are usually checked by using the finite element model and test method [12–14]. A boundary integral method is commonly used to analyze the deformation of the roll system and obtain reliable results [15]. Pawelski et al. [16] proposed a mathematical model of load distribution along the width of the strip, and Wang et al. [17] used this method and gave a calculation model of roll flattening deformation based on the plane hypothesis principle. Similarly, a similar model was designed to solve the rolling of an ultra-thin strip, where a pair of work rolls may contact outside the strip area after rolling deformation [18]. Le et al. [19] proposed an analytical method, which is well applicable to the case of excessive elastic deformation of the work roll. The Karman equation to realize the adjustment of adhesive friction was employed in the simulation of the tropical steel rolling process [20]. In addition, an analytic model for cold sandwich rolling was proposed [21].

Nowadays, more and more factories use six-high mills with a small work roll diameter to produce high strength and ultra-thin strip steel. Given the above-mentioned contributions, the effect of a small work roll diameter for the strip profile on a six-high cold rolling process has not been discussed thoroughly. A minimal cost analytical model based on the influence function method and slab method has been built, and the accuracy of the model has been verified. The rolling process of a mill with a small work roll diameter was analyzed and calculated by using the model with reference to industrial field parameters. Eventually, the important ideas emerge.

## 2. Methods and Materials

An analytical method including two methods is presented to analyze the deformation of a roll stack and strip during cold strip rolling. The influence function method is used to solve roll stack elastic deformation and the slab method is used to provide rolling pressure at points along the width of the strip for a given roll stack deformation. These two parts iterate with each other until the residual criterion is satisfied. The influence function is based on solid elastic theory to calculate the elastic deformation of the roll stack under an unknown load vector $[f]_{N\times1}$. It is obtained by the superposition of the bending term, flattening term, Poisson's ratio term, and correction term. The calculation method can adapt to the abrupt drop in diameter near the roll barrel end.

Because the cold-rolled strip steel thickness is lower, the cell needs to be subdivided in the computational model, so dividing the strip unit quantity is more. When commercial finite element software is used, the calculation time of the model usually reaches more than ten hours or even longer. The calculation time of the model established in this paper is less than two hours, which is greatly shortened compared to the finite element software. The model can be used to calculate the influence of multiple technological parameters in the cold rolling process. The precise form of each term is as follows.

### 2.1. Bending Terms

The bending deformation of the roll under the load condition can be calculated from Timoshenko beam theory. The bending influence coefficient of load $f_j$ at location $y_j$ of roll m is given by Equation (1).

$$K_m^b(y_i, y_j) = \begin{cases} \frac{y_i l_j}{k_m \pi r_m^2 E_m} + \frac{y_j y_i^2 l_j}{2 E_m I_m} - \frac{y_i^3 l_j}{6 E_m I_m} & |y_i| < |y_j|, y_i y_j > 0 \\ \frac{y_j l_j}{k_m \pi r_m^2 E_m} + \frac{y_i y_j^2 l_j}{2 E_m I_m} - \frac{y_j^3 l_j}{6 E_m I_m} & |y_i| > |y_j|, y_i y_j \geq 0 \end{cases}, \tag{1}$$

where

$$k_m = \frac{3(1 + v_m)}{7 + 6v_m - 2v_m^2}, \tag{2}$$

$$I_m = \frac{\pi r_m^4}{4},\tag{3}$$

where $K^b$ is the bending displacement coefficient under the influence of a load vector $f$, $v$ is Poisson's ratio, $A$ is the section area along the axial direction of the roll, $E$ is Young's modulus, and $I$ is the second moment.

Because of the small ratio of roll length to diameter, the roll bending calculation is affected by the Poisson effect. The influence function coefficient $K^p$ is given by Equation (4).

$$K_m^p(y_i, y_j) = \begin{cases} \frac{l_j v_m r_m^2 (L-2y_j)(L+2y_i)}{8E_m I_m L_m} - \frac{l_j v_m r_m^2 |y_j - y_i|}{2E_m I_m} & y_i - y_j \le 0 \\ \frac{l_j v_m r_m^2 (L-2y_j)(L+2y_i)}{8E_m I_m L_m} & y_i - y_j > 0 \end{cases}.\tag{4}$$

### 2.2. Flattening between Work Roll and Strip

At the same time of steel strip deformation, the work roll undergoes elastic flattening deformation due to the distributed load from the steel strip. The deformation caused by the contact between the work roll and the strip steel can be calculated by using Boussinesq's formula, as shown in Equation (5).

$$K^f(x, y, 0) = \frac{1 - v^2}{\pi E} \iint \frac{p(x', y')}{\sqrt{(x - x')^2 + (y - y')^2}} dx' dy',\tag{5}$$

where $K^f$ is the influence function coefficient for the normal displacement at point $(x, y)$ caused by the pressure distribution near point $(x', y')$.

### 2.3. Flattening between Stacked Rolls

The Hertzian contact theory can be used to explain the deformation of the contact position between rollers when the rolls are stacked and in contact. The contact arc length $c_1$ at position $x$ can be calculated by Equation (6).

$$c_1 = \sqrt{\frac{2D^* f_{y'}}{\pi E^*}},\tag{6}$$

where

$$D^* = \left[(2r_m)^{-1} + (2r_n)^{-1}\right]^{-1},\tag{7}$$

$$E^* = \left[\frac{1 - v_m^2}{E_m} + \frac{1 - v_n^2}{E_n}\right]^{-1},\tag{8}$$

where $r$ is the radius of rolls; due to distributed loads, the rolls' radiuses change and are calculated by Boussinesq's formula shown in Equation (9).

$$\begin{aligned} K_{m,n}^f(y, y', f_{y'}) = & \ (1 - v_m^2)\{b_1 \log[(m_2 + sqbm_{1,2})/(m_1 + sqbm_{1,1})] \\ & -b_2 \log[(m_2 + sqbm_{2,2})/(m_1 + sqbm_{2,1})] \\ & +2m_2 \log[(b_1 + sqbm_{1,2})/(b_2 + sqbm_{2,1})]\}/2\pi E_m c_1 \\ & -(1 + v_m)\{b_1^2/sqbr_{1,m} - b_2^2/sqbr_{2,m} \\ & +2(1 - v_m)\log[(sqbr_{1,m} + b_1)/(sqbr_{2,m} + b_2)]\}/2\pi E_m \\ & -k_m^f\{b_1(2b_1^2 + 3r_m^2)/(b_1^2 + r_m^2)^{1.5} - b_2(2b_2^2 + 3r_m^2)/(b_2^2 + r_m^2)^{1.5}\}/2\pi E_m \end{aligned}\tag{9}$$

where

$$k_m^f = \frac{7 - 25v_m - 168v_m^2 - 144v_m^3}{96(1 + v_m)},\tag{10}$$

$$b_1 = y' - y + \frac{l}{2},\tag{11}$$

$$b_2 = b_1 - l, \tag{12}$$

$$m_1 = -c_1, \tag{13}$$

$$m_1 = c_1, \tag{14}$$

$$sqbm_{1,1} = \sqrt{b_1^2 + m_1^2}, \tag{15}$$

$$sqbm_{1,2} = \sqrt{b_1^2 + m_2^2}, \tag{16}$$

$$sqbm_{2,2} = \sqrt{b_2^2 + m_2^2}, \tag{17}$$

$$sqbm_{2,1} = \sqrt{b_2^2 + m_1^2}, \tag{18}$$

$$sqbr_{1,m} = \sqrt{b_1^2 + r_m^2}, \tag{19}$$

$$sqbr_{2,m} = \sqrt{b_2^2 + r_m^2}. \tag{20}$$

An elliptic correction for the Hertzian pressure distribution has to be introduced as in Equation (21).

$$K_{m,n}^f\left(y, y', f_{y'}\right) = \begin{cases} K_{m,n}^f + \frac{(1-v_m^2)(2\log 2 - 1)}{\pi E_m} & y = y' \\ 0 & y \neq j' \end{cases}. \tag{21}$$

*2.4. Rolling Pressure Resulting from Strip Elastic–Plastic Deformation*

The rolling pressure distribution along the width of the strip can be calculated using the slab method [22] when the displacement of each strip along the width is calculated and the forward and backward tension is given. Equations for the equilibrium of forces applied to a slab element are shown in Equations (22) and (23).

$$h\frac{d\sigma_{xx}}{dh} + (\sigma_{xx} + p)\frac{dh}{dx} + 2q_x = 0, \tag{22}$$

$$h\frac{d\tau_{yx}}{dx} + 2q_y = 0. \tag{23}$$

Among them, $x$, $y$, and $z$ represent the rolling direction, strip width direction, and thickness direction, respectively, $p$ is the rolling pressure, $\sigma_{xx}$ is the tension stress with an initial value of back tension, and $t$ is the thickness. $q$ is the shear stress, where its $x$ and $y$ components are calculated by Equations (24) and (25).

$$q_x = \pm\mu p \frac{dV}{\sqrt{dV^2 + dU^2}}, \tag{24}$$

$$q_y = \pm\mu p \frac{dU}{\sqrt{dV^2 + dU^2}}, \tag{25}$$

where $\mu$ is the friction coefficient, and the positive sign is for the backward slip zone. $V$ and $U$ can be calculated by Equations (26) and (27).

$$dV = \int_{x_n}^{x} d\varepsilon_x dx, \tag{26}$$

$$dU = \int_{0}^{y} d\varepsilon_y dy, \tag{27}$$

$$d\varepsilon_z = \ln\frac{h + dh}{h}, \tag{28}$$

$$d\varepsilon_x = -\frac{1}{1+G}d\varepsilon_z, \tag{29}$$

$$d\varepsilon_y = -\frac{G}{1+G}d\varepsilon_z, \tag{30}$$

where $G$ is the metal transverse flow factor inside the rolling region. Based on finite element model (FEM) calculation, the transverse flow factor is dependent on yield stress, friction coefficient, reduction ratio, and strip thickness, and it can be calculated by Equation (31).

$$G(y) = \begin{cases} \frac{G_{\max}}{(b^2/4 - y_0^2)}\left(y^2 - y_0^2\right) & |y| \in [y_0, b/2] \\ 0 & |y| \notin [y_0, b/2] \end{cases}, \tag{31}$$

where $b$ is the strip width, $y_0$ is the width position where metal flows transversely, and $G_{max}$ is the maximum transverse flow factor. They are calculated as follows:

$$G_{\max} = 0.05 \cdot k_S \cdot k_\mu^{\max} \cdot k_t^{\max}, \tag{32}$$

$$k_S = 0.009(Y_S - 100) + 1, \tag{33}$$

$$k_\mu^{\max} = 6(\mu - 0.05) + 1, \tag{34}$$

$$k_t^{\max} = [0.0037(70 - \sigma_f) + 1] \cdot [0.0051(50 - \sigma_b) + 1], \tag{35}$$

$$y_0 = 30 \cdot k_h \cdot k_r \cdot k_\mu^{y_0}, \tag{36}$$

$$k_h = 0.21\left(h_{\text{entry}} - 1.2\right) + 1, \tag{37}$$

$$k_r = 2(re - 5) + 1, \tag{38}$$

$$k_\mu^{y_0} = 5.13(0.18 - \mu) + 1, \tag{39}$$

where $r_e$ is the reduction ratio and $Y_S$ is the yield stress. When thickness $t$ is smaller than the entry thickness, the strip undergoes elastic deformation and the resulting rolling pressure is calculated by Equation (40).

$$\frac{\mathrm{d}p}{\mathrm{d}x} = -\left(1 - \frac{v}{1-v}\frac{G}{1+G}\right)\frac{E}{h}\frac{\mathrm{d}h}{\mathrm{d}x} + v\frac{2q_x}{h}. \tag{40}$$

When the sum of the pressure and tension stress becomes larger than yield stress $Y_S$, plastic deformation happens and the rolling pressure is calculated by Equation (41).

$$\sigma_{xx} + p = \sqrt{Y_S^2 - 4\tau_{xy}^2}. \tag{41}$$

When the shear stress is in parallel with the slope of the work roll, the strip thickness tends to re-increase inside the rolling region. The pressure should be regularized to ensure the strip moves with the same speed longitudinally as the work roll. The pressure can be obtained by Equation (42).

$$\frac{\mathrm{d}p}{\mathrm{d}x} = -\frac{C_1 E}{t}\frac{\mathrm{d}h}{\mathrm{d}x} + \frac{\mathrm{d}Y_S}{\mathrm{d}x} \quad q > -\left(\frac{C_1 E}{2}\right)\frac{\mathrm{d}h}{\mathrm{d}x}. \tag{42}$$

$C_1$ is placed at 1.8. When the slope of the work roll is positive, elastic relaxing happens again. A 4th order Runge Kutta method is utilized to integrate these 1st order differential equations (ODEs). An iterative campaign has to be utilized in order to find the neutral point where the shear stress changes sign. The neutral point iteration stops if the exit tension stress is converged to the imposed exit tension. The work hardening law of the non-oriented silicon strip is simplified as in Equation (43).

$$Y_S = (440 + 140\varepsilon_{zz})\left(1 - 0.45\mathrm{e}^{-25\varepsilon_{zz}}\right) - 25. \tag{43}$$

*2.5. Model Calculation Flow*

In the process of cold rolling, the three-dimensional deformation of the strip edge metal is complicated. The forming of the strip edge shape is obtained by coupling elastic deformation of the roll and three-dimensional deformation of the strip. The rolling pressure distribution of the strip along the width is uneven, which will cause uneven elastic deformation of the roll. The uneven deformation of the roll will affect the transverse flow of the metal at the edge of the strip, which will cause the uneven distribution of tensile stress after the strip rolling and further affect the distribution of rolling pressure. This is a cyclic iteration of the impact and calculation relationship, as shown in Figure 2. According to

the interaction between the roll and the strip during rolling, a flow chart of the mathematical calculation model for the roll and the strip is illustrated in Figure 3.

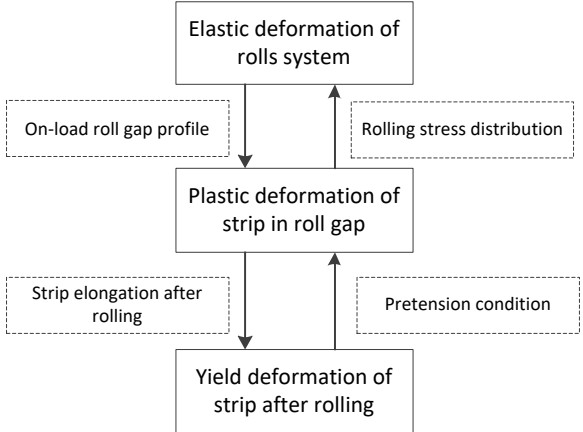

**Figure 2.** Roll and strip influence on each other during and after rolling.

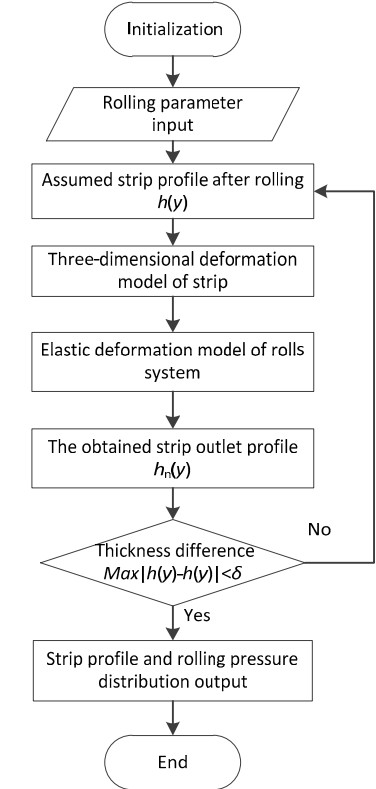

**Figure 3.** Roll and strip calculation model flow chart.

### 2.6. Model Parameters

The parameters employed in the calculation model are derived from a 1420 mm Universal Crown Mill (1420mmUCM). This mill is generally equipped with working roll diameters ranging from 340 to 360 mm. After the modification of equipment parts, work rolls with a diameter of 230–260 mm can be equipped. It is a known fact that rolling mills with a smaller-diameter work roll can produce steel strips with higher strength and lower thickness. Many factories have two rolling mills to accommodate the variety of silicon steel products.

This paper analyzes the shape control characteristics of a silicon steel strip when two different work rolls are used. The model parameters are the same except the diameter of the work roll. Geometric

parameters of the selected mill are given in Table 1. The selection of the rolling process is shown in Table 2. The rolled strip is a non-oriented silicon steel strip.

**Table 1.** 1420mmUCM reversible mill geometry parameters.

| Parameters | Value |
|---|---|
| Backup roll length/mm | 1420 |
| Backup roll neck diameter/mm | 690 |
| Work roll length/mm | 1420 |
| The distance between two bending points of work roll/mm | 2150 |
| Intermediate roll length/mm | 1445 (Chamfer 50) |
| The distance between two bending points of intermediate roll/mm | 2800 |

**Table 2.** 1420mmUCM reversible mill rolling parameters.

| Parameters | Value |
|---|---|
| Entry/exit thickness/mm | 2.200/1.360 |
| Reduction/% | 32.0 |
| Work roll bending force/kN | 120 |
| Intermediate roll bending force/kN | 150 |
| Before/after tension/MPa | 20/118 |
| Intermediate roll shifting value/mm | 20 |
| Strip width/mm | 1230 |

With the UCM mill as the research object, when rolling the silicon steel strip, it is usually equipped with a unique roll shape curve of the work roll to control the profile of the silicon steel strip. Therefore, when the influence of a small work roll diameter on the shape of the silicon steel strip is studied, the work roll with a roll shape curve and the one without the curve is selected to study. In this way, the control mechanism of reducing the work roll diameter on the shape of the strip can be studied, and the comprehensive control effect of work roll diameter and work roll curve can be studied. The roll system configuration is shown in Figure 4 and the work roll shape curve is shown in Figure 5, which has been successfully implemented in many industrial sites and has effective control on the transverse thickness difference of the silicon steel strip.

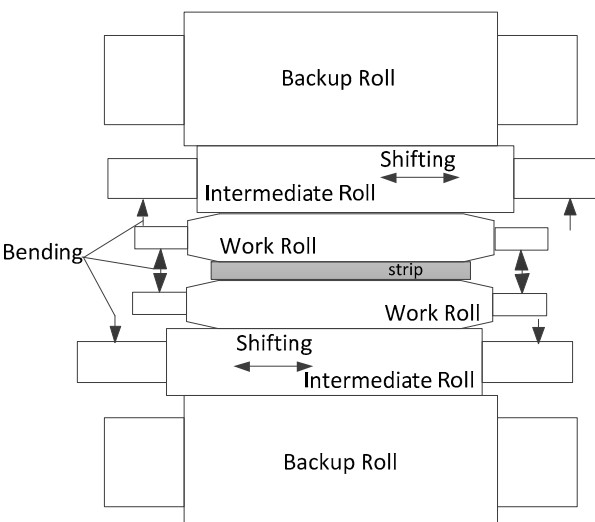

**Figure 4.** Rolls stack and strip system.

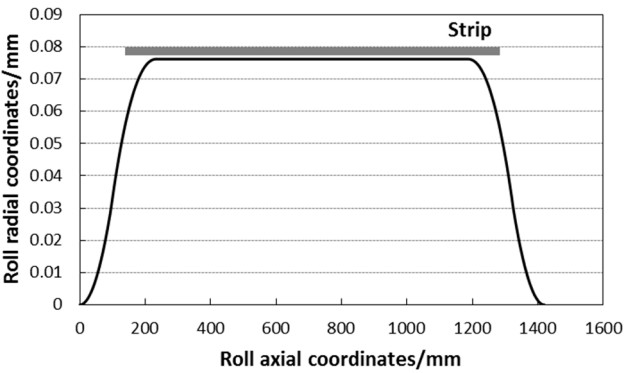

**Figure 5.** Work roll shape curve.

## 2.7. Model Calculation Accuracy Verification

In order to verify the calculation accuracy of the established calculation model, the measurement results of industrial strip samples are compared in this paper. The samples were selected from the first rolling of the non-oriented silicon steel named 50SW800. The shape profile was measured by an automatic thickness gauge with a grating sensor. The resolution of the instrument was 1 μm and the error range was ±1 μm. The section profile of the strip between rolling passes was measured in practice. Then, the model calculated value of the first rolling step was compared to the measured value, as shown in Figure 6. The strip indicated in the figure is a half-width edge profile.

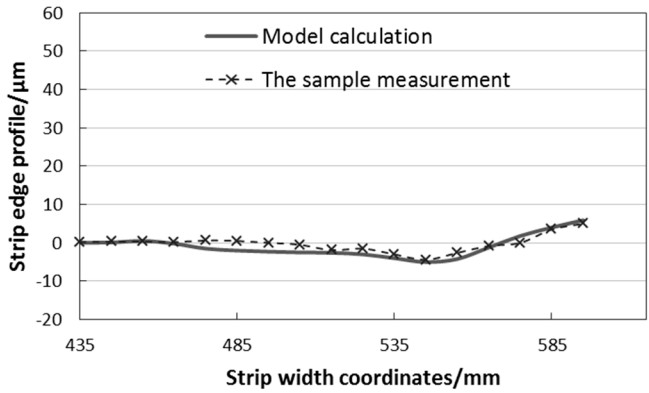

**Figure 6.** Comparison of model calculated value of strip edge profile with sample measurement.

As can be seen from Figure 6, the area with the error of section calculation results of less than 2 μm can reach more than 90% for the 15–180 mm region in the edge drop region. The error source of the strip edge region may be the calculated friction and deformation state in the edge region, and the actual value of the transverse flow factor has a deviation from the theoretical value, which has a limited effect on the accuracy of strip section calculation. Therefore, the accuracy of the model can be guaranteed.

## 3. Analysis of Calculation Results

The coupling calculation method is used to calculate the deformation of the roll and the strip of the given parameters. The bending deformation of the work roll, the flattening deformation of the work roll, and the strip contact and section shape of the strip are extracted from the calculation results. The effects of the reduction of work roll diameter on each part are analyzed. At the same time, the effect of roll diameter on the edge drop control of the curve was calculated by setting two kinds of work roll curve forms. The exact analysis is as follows.

*3.1. Calculation of Roll Flattening Deformation*

The mechanism of edge drop improvement of a small work roll radius is further analyzed. It can be found that the elastic flattening amount on the surface of a large-diameter roll obviously exceeds that of a small-diameter work roll. As shown in Figure 7, the difference in the flattening amount of the roll along the width of the strip calculated is 16 μm. As the contact position with the edge of the strip is the connection section of the flattening zone and the non-flattening zone, a large amount of flattening will cause the rise in the edge drop of the strip. This is particularly true within a 200 mm distance from the edge of the strip.

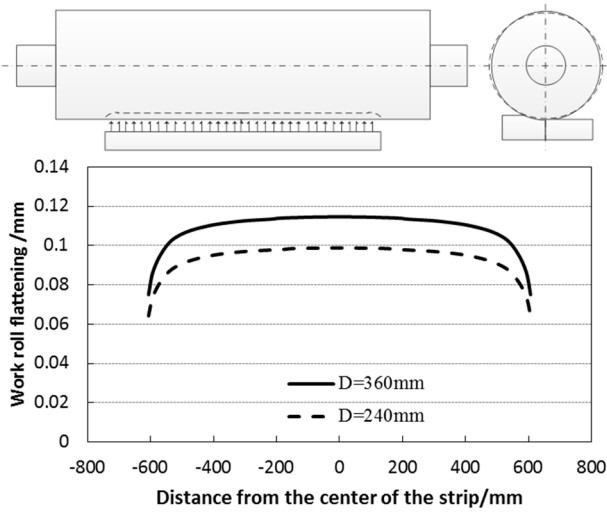

**Figure 7.** Elastic flattening of work roll with different diameters.

*3.2. Calculation of Cross-Sectional Profile of Strip*

The calculation results with two kinds of work roll diameter in the first rolling pass are given in Figure 8a where a horizontal line is used for the work roll curve (no special curve). It can be found that the strip shape with a smaller work roll radius is flatter and the crown is smaller. There is no obvious transition zone from the middle straight segment to the edge drop zone. The edge drop within the 60 mm area of the strip edge is 47.9 μm. However, for the mill with the large work roll radius, the strip edge drop is 54.4 μm and the overall strip crown is larger. The reduction of roll diameter can reduce the edge drop by 6.5 μm.

The calculation results with two kinds of work roll diameter in the first rolling pass are given by Figure 8b when the work roll curve is as shown in Figure 5. It can be seen that the edge of the strip shows a more obvious increase in thickness when the small roll diameter mill is used, the edge drop of the strip is −61.45 μm, and the crown of the strip shows a negative value. However, for the mill with a large work roll radius, the increase in edge thickness is relatively small. The strip edge drop is −39.54 μm. The edge drop decreases by 21.91 μm. It shows that for the same work roll shape curve, the edge drop control ability of the small-roll radius mill is higher than that of a large one. For a mill with a large work roll radius, it is necessary to adopt the shape of the work roll with stronger control ability to improve the edge drop. This will inevitably result in more complex wave problems.

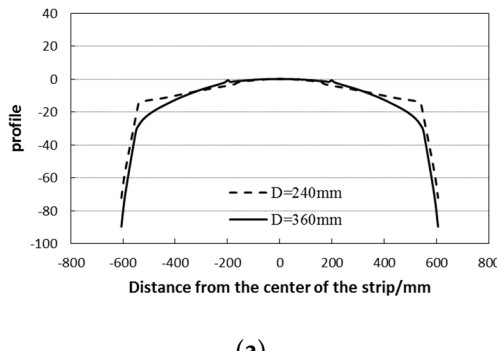 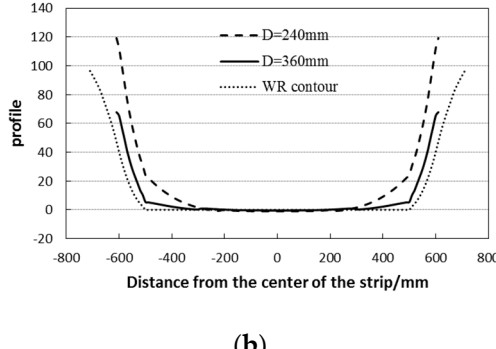

(**a**)                       (**b**)

**Figure 8.** Comparison of profile calculation results with different work roll diameters: (**a**) No work roll contour; (**b**) with work roll contour.

The reason the small roll diameter is more advantageous to control the edge drop of the strip is analyzed only through the integrated model of the roll system and strip. It is necessary to track the control effect of the edge drop of different roll diameters through industrial field tests.

This part focuses on the analysis of the small roll diameter for the improvement principle of the silicon steel shape, but the work roll diameter can be reduced indefinitely. This paper only discusses the work roll of an UCM six-high mill shape improvement. If the roll diameter becomes smaller, the roll will produce a larger transverse deflection along the rolling direction, and the research direction will be extended to more rolling mills, such as a 12 high mill or 20 high mill.

## 4. Industrial Application

The Maanshan Iron & Steel Company Ltd. in Maanshan city, Anhui province, China, is equipped with two 1420 mm six-high reversible cold rolling mills to accommodate the production of different types of silicon steel strip, as shown in Figure 9. The allowable range of work roll diameters of the two mills is different. It is 230 to 260 mm for the No.1 mill and 340 to 360 mm for the No.2 mill. The No.1 mill has been modified and can be equipped with small-diameter work rolls. The actual production data of non-oriented silicon steel in Table 3 serve to illustrate the influence of the work roll diameter on the strip shape. The chemical composition of selected samples is shown in Table 4, and the magnetic and mechanical properties requirements are shown in Table 5. The shape profile was also measured by an automatic thickness gauge with a grating sensor.

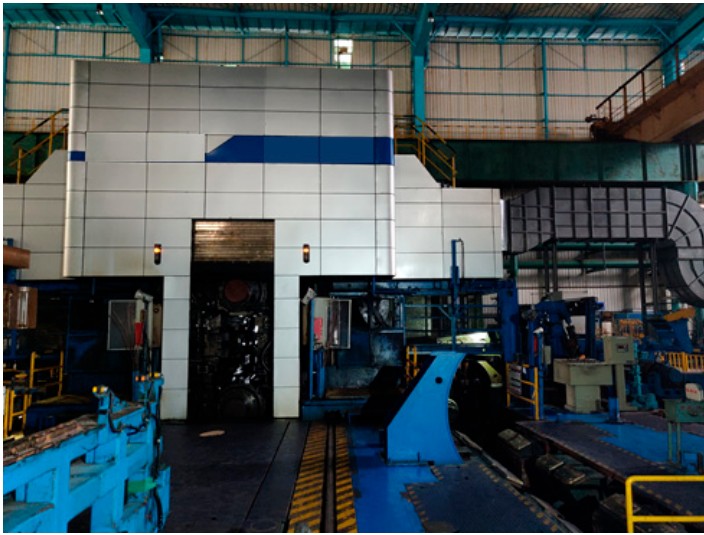

**Figure 9.** Schematic diagram of six-high reversible cold mill production.

**Table 3.** Parameters of non-oriented silicon steel in cold rolling.

| Steel Grade | Width | Entry/Exit Thickness |
|---|---|---|
| 50SW800 | 1230 mm | 2.5/0.5 mm |

**Table 4.** Chemical composition of non-oriented silicon steel (wt.%).

| C | Si | Mn | Al | P | S |
|---|---|---|---|---|---|
| 0.003 | 0.85 | 0.25 | 0.33 | 0.03 | 0.0045 |

**Table 5.** Magnetic and mechanical properties of non-oriented silicon steels.

| Iron Loss $P_{15/50}$ ($\leq$W/kg) | Magnetic Induction $B_{50}$ ($\geq$T) | Number of Reverse Bending in Rolling Direction | Residual Curvature (mm) |
|---|---|---|---|
| 8 | 1.55 | $\geq$2 | $\leq$35 |

After six months of industrial field experiment tracking, it was found that the hit rate of the No.1 mill with the transverse thickness difference of the strip of less than 10 μm remained stable at about 95%, while the hit rate of the No.2 mill greatly fluctuated, as shown in Figure 10a. At the same time, the probability of the strip transverse thickness difference of the No.1 mill of less than 5 μm could reach 50%, which is significantly higher than that of the No.2 mill, as shown in Figure 10b.

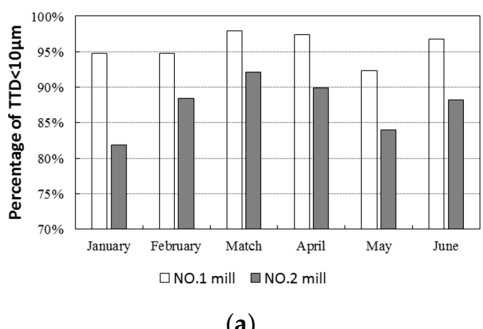 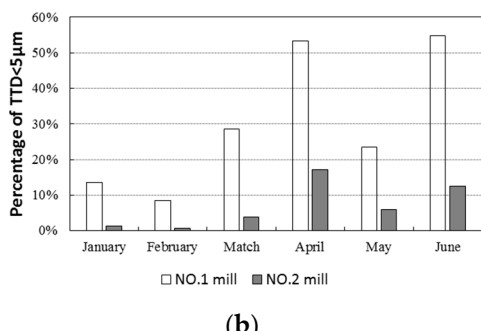

(**a**)　　　　　　　　　　　　　　　　　　　　(**b**)

**Figure 10.** Statistics of transverse thickness difference (TTD) of strip produced by two rolling mills: (**a**) Less than 10 μm; (**b**) less than 5 μm.

The typical cross-sectional profile of the non-oriented silicon steel sample rolled by the two mills is illustrated in Figure 11. At this time, the same roll shape curve is utilized on the work rolls of the two mills. It can be observed in the figure that the thickness change of the strip edge affected by work roll shape is very little. The difference in transverse thickness between the two templates is mainly reflected in the middle region, which is consistent with the simulation results in Section 3.1, that is, the elastic flattening caused by different work roll diameters is different, thus affecting the section shape of the strip.

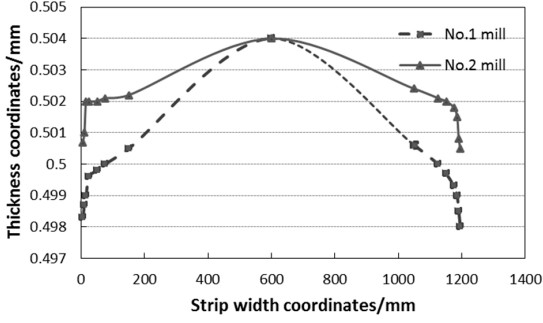

**Figure 11.** Typical section shape comparison of strip produced by different mills.

In the UCM mill production process, in order to improve the lateral thickness difference of silicon steel, the work roll shape curve often has to be optimized. Figure 12 shows the end curves of two twin taper work rolls. Two schemes were tested in the No.1 mill and No.2 mill, and a large number of transverse thickness difference data boxes measured for silicon steel after rolling are shown in Figure 13. The number in the abscissa represents the mill number, and the letter represents the curve number. The average values of the four groups of data were extracted, as shown in Table 6. It can be found that the transverse thickness difference of the strip decreases by 2.3 μm for the No.1 mill and 1.1 μm for the No.2 mill. This also indicates that the tapered work roll curve of the rolling mill with small work roll diameter can further enhance the control effect on the transverse thickness difference of the strip steel, which is in agreement with the conclusion in Section 3.2.

Of course, the height of the curve at the edge of the tapered work roll cannot be increased without limit, which will cause excessive edge stress and local wave problems. The design of a roll shape curve needs to be optimized by integrating multiple plate shape targets.

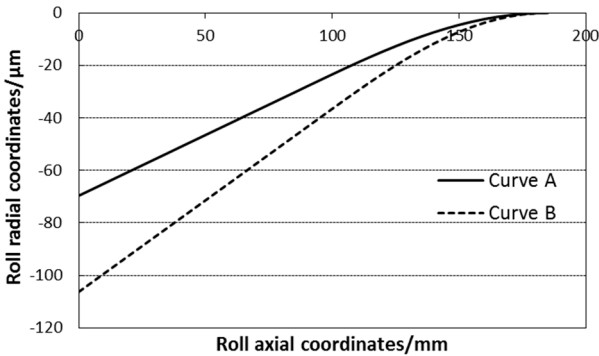

**Figure 12.** Comparison of work roll shape curves.

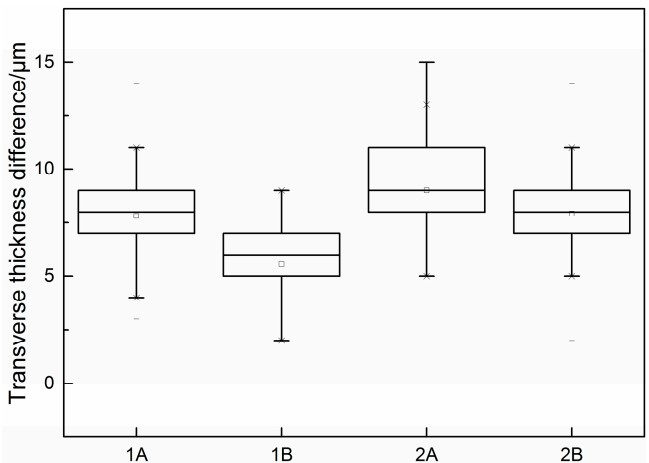

**Figure 13.** Box diagram of transverse thickness difference of strip under different parameters.

**Table 6.** Statistics of transverse thickness difference of strip.

| Work Roll Contour | No.1 mill | No.2 mill |
|---|---|---|
| Curve A | 7.8 μm | 9 μm |
| Curve B | 5.5 μm | 7.9 μm |

## 5. Conclusions

(1) A roll-strip coupling model is used to simulate the rolling process of a six-high mill. The influence function method is used to solve roll stack elastic deformation and the slab method is used to provide rolling pressure at points along the width of the strip for a given roll stack deformation.

The model with low computational cost and high enough accuracy can be used to calculate the strip deformation under various technological parameters during cold rolling.

(2) The mechanism of improving the shape of the work roll by reducing its diameter is analyzed and two important conclusions have been obtained. Smaller elastic flattening in cold rolling can be produced for work rolls with a small diameter. Thus, the profile of the strip section is improved. The shape control ability of the taper work roll can be further improved for the mill with a small work roll diameter when it is combined with the taper work roll technology.

(3) At present, this technology has been implemented in the industrial field and can effectively control the lateral dimension accuracy of silicon steel with the work roll profile curve, which greatly improves the shape control ability of a UCM mill.

**Author Contributions:** H.H. and Q.Y. conceived and designed the work roll shape curve; X.F. carried out the simulation work; H.H. performed the industrial experiments; X.W. and J.S. analyzed the data; H.H. wrote the paper. All authors have read and agreed to the published version of the manuscript.

**Funding:** This research was funded by Fundamental Research Funds for the Central Universities (grant No. FRF-TP-19-002A3) and the APC was funded by Fundamental Research Funds for the Central Universities (Grant No. FRF-TP-19-002A3).

**Acknowledgments:** The authors would like to thank the National Natural Science Foundation of China (Grant No. 51975043), Fundamental Research Funds for the Central Universities (Grant No. FRF-TP-19-002A3), and Beijing Natural Science Foundation (3182026) for the support to this research.

**Conflicts of Interest:** The founding sponsors had no role in the design of the study; in the collection, analyses, or interpretation of data; in the writing of the manuscript, and in the decision to publish the results.

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
