# Peer review of "Silicon Steel Strip Profile Control Technology for Six-High Cold Rolling Mill with Small Work Roll Radius"

_metals, doi:10.3390/met10030401_

Round 1

Reviewer 1 Report

In the present study authors investigated the “Silicon steel strip profile control technology for 2 6-high cold rolling mill with small work roll radius”. Authors mainly solve theoretically setting of optimal parameters for cold rolling process of the silicon steel strip. It is a calculation of model parameters and their analysis for industrial use. I think that the research is interesting and useful for people working on cold rolling.

 Questions and comments:

        1) Please indicate the material properties and detail chemical composition of silicon steel.

(Material and magnetic properties, and also rolling condition would be different for silicon steel with silicon content 2 or 6 mass percent. Silicon steel with 6wt% of silicon is too hard and brittle for commonly rolling.)

2) Figure 4 contains lots of abbreviation without explanation of meaning in text or figure description. Rolled strip is mark as rectangle with stripes?

3   Line 37 – Please specific what kind of percent was mentioned – atomic or mass? The difference is crucial.

3) In some part, typographical character is needs to correct:

At the end of sentence leave a blank space after dot. (Line no. 81, 85, 86)

Correct form of references no. 3, 9, 10.

Reviewer 2 Report

The paper concerns technology of 6 - high cold rolling mill with small work roll radius. The authors presented analytical and experimental solutions that can increase the quality of final products. Some elements of the manuscript need to be revised and expanded:

1. The rolling process significantly affects the texture and structure of the domain walls of grain oriented or non-oriented silicon steel sheets. It seems to me that one could mention this in the analysis of the state of knowledge, i.e. what is the significance of this process in forming the specific structure of texture of the material and, as a result, magnetic properties.

2. The main aim of the paper and its significance against the background of the current state of knowledge should be more detailed described. 

3. The experiment presented in section 2.7 and 4 could be more detailed described. There is no description of the test conditions, measuring sensors, their accuracy, etc. 

Line 314: "It can be found that the transverse thickness difference of the strip decreases by 1.08um" -  the measurement is very accurate. Accuracy of sensors and error margin should be described. 

4. How the three-dimensional plastic deformation of the strip was measured? 

5. The material used for testing should be described in more detail. Chemical composition, mechanical properties should be presented

The paper should be formatted. Some sentences are complicated an may be not understood by readers. For example "He developed contours that differ significantly from each other, even though they all concentrated on the design of tapered work roll contour" - I think citation is missed. 

Fig. 8 - description is missing at the same page.

Please check the journalist's requirements about citation style. Usually citation is given as follows:  Pawelski et al. [16].....    
